# Operation Method of PV–Battery Hybrid Systems for Peak Shaving and Estimation of PV Generation

Kun-Yik Jo [1] and Seok-Il Go [2],*

1    Department of Electrical Engineering, Chonnam National University, Gwangju 61186, Republic of Korea
2    Department of Electrical Engineering, Honam University, Gwangju 62399, Republic of Korea
*    Correspondence: riseisgood@honam.ac.kr

**Abstract:** Photovoltaic (PV)–battery hybrid systems, which are composed of PV arrays, batteries, and bidirectional inverters, can level the loads of traditional utility grids. Their objective is to supply predetermined active and reactive power to the power grid. This paper presents an operation method for PV–battery hybrid systems by estimating PV generation. Using the PV installation information, the maximum PV generation on a clear day was predicted and compared with historical data. The PV generation was estimated using historical data from 2007 to 2010. The method aims to reduce the peak load of the power system using the estimated load and PV generation of the next day. With the given weather information and load pattern for the next day, the charge and discharge set points of the battery can be determined by considering the initial SoC (State of Charge) and capacity of the battery. To compensate for the estimation error of the load and PV output, an operational margin was considered. This method can maximize system operation efficiency by fully utilizing the battery. The effectiveness of the operation method was validated through simulation studies. It was confirmed that the peak load could be reduced by 30% using the proposed algorithm.

**Keywords:** photovoltaic; battery; battery energy storage system (BESS); PV–battery hybrid system; solar power generation estimation; peak load reduction

## 1. Introduction

Photovoltaic (PV) generation involves the conversion of solar energy to electric energy using photovoltaic cells. Owing to its advantages, such as abundant resources, easy exploitation, cleanliness, and renewable characteristics, PV generation is developing rapidly as a renewable energy source. However, the disadvantage of PV generation is that it is intermittent owing to its dependence on weather conditions. Thus, energy storage elements are necessary to obtain a stable and reliable system output from a PV generation system for various load conditions and to improve both the steady-state and dynamic behaviors of the PV generation system [1].

Renewable energy sources, including solar photovoltaic systems and wind power (WT), are key technologies for carbon-free energy production [2]. Fossil-fuel-based power plants can provide electrical energy to the market because of their generally lower production costs compared with PV and WT [3]. However, they are a source of carbon dioxide, and the cost of electricity generation is expected to increase as the cost of emission allowances rises due to the implementation of the EU Emissions Trading Scheme (EU ETS) [4]. The intermittent nature of renewable energy generators based on solar and wind energy resources makes effective power system control difficult, and the use of these energy sources in the shared energy mix is expected to increase steadily over the next few decades [5]. Lithium-ion battery research is receiving considerable attention worldwide. Batteries are becoming safer and cheaper, and the technology needed to use them in power distribution systems is getting a lot of attention [6,7].

Grid-tied PVs with battery energy systems have been widely studied to simulate and quantify the optimal benefits of deploying such systems. Some of the system aspects studied



include the simulation and optimization of PV systems based on energy price and demand forecasts [8–10]. Recently, the batteries used in grid-connected PV systems have received considerable attention, especially with regard to their suitability and usage time [11].

Despite high installation costs, domestic solar PV has a high adoption rate which is driven by energy policies, such as conformity schemes in Europe and other parts of the world [12–14]. The adoption of batteries in renewable energy systems with high fluctuations in distributed energy resources can relieve the output fluctuations caused by specific electricity demands or grid-connected distributed energy systems [15,16]. An important issue in this context is to justify the need for battery storage systems in electrical networks. The peak load demand of power systems is increasing, and the high share of distributed energy resources creates a mismatch between generation and demand [17]. Thus, the utilization of power generation, transmission, and distribution infrastructure is not suitable for power system operation.

Utility operators can leverage battery storage with PV systems to maximize the use of existing network capacity and defer network investments. Therefore, the capacity of home electricity customers to provide an effective response to dynamic electricity prices will be increasingly valuable for integrating the high penetration of distributed energy resources such as PV into future electricity networks. In [10], the effects of active demand-side management and battery storage systems on self-consumption were investigated. The relationship between the electrical energy flow and battery storage capacity has been shown to be an important determinant. A study [18] investigated the viability of suitable systems to enhance the development of renewable energy technologies. Furthermore, batteries have been used in the campus microgrid field as well as in many other fields [19,20].

A battery energy storage system (BESS) can be integrated into a PV generation system to form a PV–battery hybrid system, which can be more stable and reliable. A PV–battery hybrid system is composed of a PV array, battery, power electronic converters, controllers, and utility grid [21]. A PV–BESS hybrid system can mitigate the intermittence of PV by controlling the charge and discharge of the ESS and contributes an auxiliary service to the grid through peak load shaving in the utility grid [22]. Recently, interest has increased in BESS-based peak shaving, which requires a scheduling strategy based on PV prediction. The purpose of this system is to supply predetermined or controllable active and reactive power to the grid. In the industrial field, the linear optimization problems based on demand and billing systems in industrial applications have allowed BESS to pay back investment costs in a short period of time [23]. In the distribution system, the load peak shaving problem has been solved by utilizing BESS in the planning stage. It was defined as an optimization problem considering time of use (TOU) and load probability distribution uncertainty, which mitigated the operating cost in the distribution system [24,25]. In most PV–BESS systems, the problem of load peak shaving brings economic benefits, but the uncertainty problem of PV must be considered. We have accounted for and solved the uncertainty problem through statistical methods [26]. In addition, the prediction problem has been solved through artificial intelligence technologies such as deep learning and machine learning [27], and through this, the optimal ESS operation plan for load peak reduction has been derived [28].

The conventional method proposed an ESS operation method with improved performance by applying artificial intelligence and stochastic statistical techniques. However, this method increases the computational burden of the system and is difficult to apply to small systems. In addition, many factors related to prediction are required and a large amount of memory is required to store them. The conventional methods mentioned above do not present methods for application to actual systems, but focus on optimization problems in consideration of cost. In addition, conventional BESS operation methods for peak shaving suggest a complex method for stabilizing the PV output. This paper presents a method for operating a PV cell system using solar power generation estimation to improve and simplify the PV operation efficiency. The maximum solar power output for each time period was calculated by the mathematical modeling of the solar power generation output.

A method for estimating solar energy was proposed by comparing the maximum amount of power generation and the actual solar output according to the weather. The purpose of this method is to reduce the peak load of the power system using the estimated load and estimated solar power generation on the next day. Given the weather information and the load pattern of the next day, the battery's initial SoC (State of Charge) and capacity can be considered to determine the battery's charge/discharge set point. The operating margin was considered to compensate for the estimation errors of the load and PV output. This method can make the most of the battery and maximize the operating efficiency of the system. The contributions of this paper are as follows:

(1) The PV prediction model was designed based on mathematical modeling and cumulative data analysis. Historical data is classified as PV output data according to the weather and expressed as a generation rate, and other factors are not considered. The generation rate can predict the maximum output of the PV through a simple calculation.

(2) Battery charge/discharge settings are determined based on predicted weather information and load patterns. In addition, the output error of load and PV can be compensated for by considering an operating margin. It has been validated as providing improved performance through simulation.

(3) The method proposed in this paper utilized data from an actual PV–BESS system. The data of the installed PV was utilized, which is suitable for validating the simulation.

This paper discusses the estimation of PV generation and presents an operation method that reduces the peak load. If the weather information and load pattern of the next day are provided, the peak load can be reduced by controlling the system output. The effectiveness of the operation method was validated through simulation studies. It was confirmed that the peak load could be reduced by 30% using the proposed algorithm.

## 2. PV–Battery Hybrid Systems

Figure 1 shows the configuration of a PV–battery hybrid system. The PV system was first introduced, followed by the BESS [29].

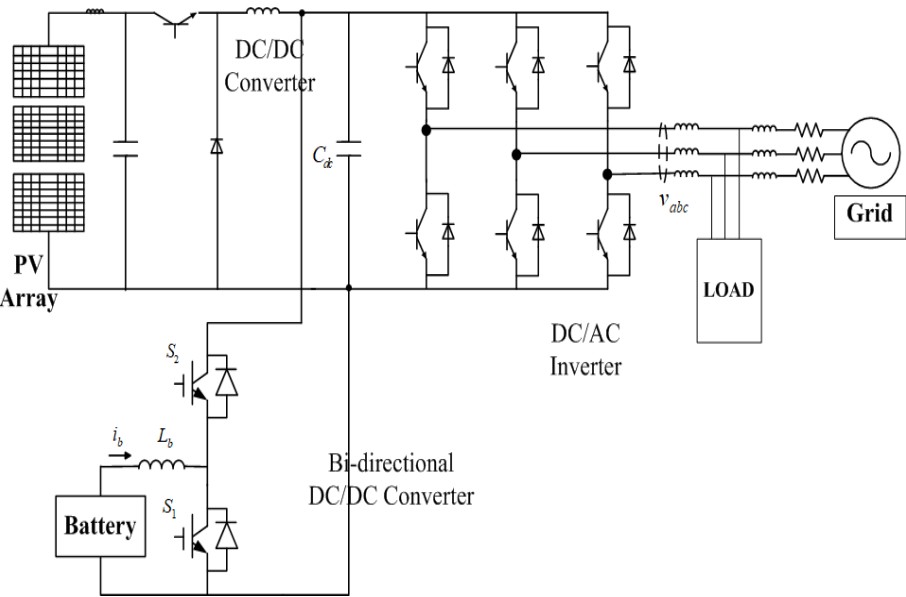

**Figure 1.** Configuration of a PV–Battery Hybrid System.

The PV cell array and batteries are linked to the common DC point via a DC/DC converter and then interconnected to the AC grid via a DC/AC inverter. The battery energy storage can charge and discharge to balance the power between PV generation and load demand. The PV system, BESS, and inverter each has an independent control objective, and, by controlling each part, the entire system operates safely [30].

The PV cell is a DC electric source. When the PV system is connected to the grid, power electronics are used to convert DC to AC power. Meanwhile, to improve the efficiency of the PV generation system, the PV array should be controlled to generate maximum power in a particular environment. For a two-stage PV system, maximum power point tracking is realized by controlling the DC/DC converter [31].

A BESS is composed of a battery, bidirectional DC/DC converter, and control system. The system should be able to operate in two directions: the battery can store the extra energy by charging and supply the required energy to loads by discharging [32].

In this study, the BESS is connected to the DC bus through a bidirectional DC/DC converter. The battery serves as a power source to meet the load demands, which cannot be fully satisfied by the PV system, particularly during solar fluctuations. Therefore, a battery is designed to compensate for the PV system.

The PV cell array and batteries are connected to the AC grid via a common DC/AC inverter. The objective of the inverter is to control the system output power regardless of the magnitude of the PV power output. A vector-control approach is used, with a reference frame oriented along the power system voltage vector position, enabling independent control of the active and reactive power flowing between the power system and inverter system. The converter system is current controlled, with the direct-axis current used to regulate the system output power and the quadrature-axis current component used to control the reactive power [33].

## 3. Estimation of PV Generation

A method for predicting the solar power output on the next day was proposed. Furthermore, a method for determining the maximum solar power output through the calculation and determination of the solar power output according to the weather was presented.

On average, pointing the collector towards the equator and tilting it at an angle equal to the latitude is a good rule of thumb for annual performance. To accentuate the winter collection, a slightly higher angle can be used, and vice versa for summer efficiency. Drawing the Earth/solar system, it is easy to determine the main sun angle, that is, the altitude angle $\beta_N$ of the sun at noon. The elevation angle is the angle between the Sun and the local horizon below the Sun. In Figure 2, we examine the following relationship, as given in [34]:

$$\beta_N = 90° - L + \delta \tag{1}$$

$$sin\beta_N = cosL + cos\delta + sinLsin\delta \tag{2}$$

$$H = 15(12 - CT) \tag{3}$$

$$CT = 4(local\ time\ meridian - local\ longitude) - E \tag{4}$$

where $H$ is the hour angle at sunrise, $CT$ is the clock time, and $E$ is the equation of time.

The position of the sun at any time of the day can be described by the altitude $\beta$ and azimuth $\varnothing_S$ angles, as shown in Figure 3. Generally of solar activity, the azimuth in the northern hemisphere is measured in degrees from the south, whereas in the southern hemisphere, it is measured relative to true north. By convention, the azimuth is positive in the morning when the sun is to the east and negative in the afternoon when the sun is to the west.

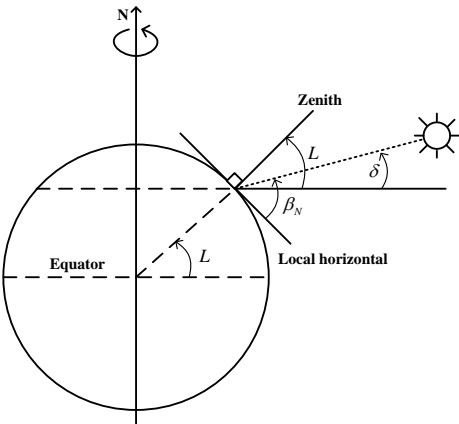

**Figure 2.** Altitude angle of the sun at solar noon.

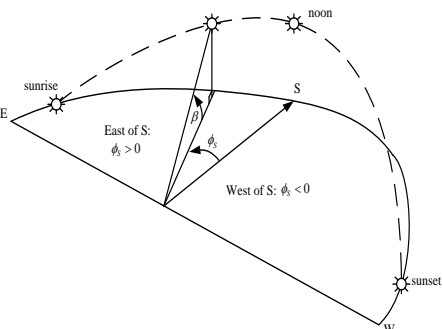

**Figure 3.** The Sun's position can be described by its altitude $\beta$ and azimuth $\varnothing_S$ angle.

The angle of incidence $\theta$ is a function of the PV cell orientation and elevation and the azimuth of the sun at a particular time. Figure 4 shows these important angles. The solar collector is tilted at angle $\Sigma$ and faces in the direction described by the azimuth $\varnothing_C$. Figure 4 illustrates the collector's azimuth angle $\varnothing_C$ and tilt angle $\Sigma$ along with the solar azimuth angle $\varnothing_S$ and altitude angle $\beta$.

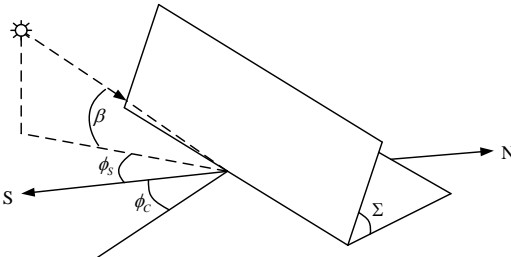

**Figure 4.** Illustration of the PV panel angles.

The solar flux striking the PV panel is a combination of direct beam radiation transmitted through the atmosphere to the receiver in a straight line, diffuse radiation scattered by molecules and aerosols in the atmosphere, and ground or reflected radiation, as shown in Figure 5. Equations (10)–(12) are the direct beam, diffuse, and reflected isolation, respectively. Equations (1)–(9) express each component of Equations (10)–(12) [34].

$$m = \frac{1}{sin\beta} \tag{5}$$

$$k = 0.174 + 0.035 sin\left[\frac{360}{365}(n - 100)\right] \quad (6)$$

$$A = 1160 + 75 sin\left[\frac{360}{365}(n - 275)\right] \quad (7)$$

$$I_B = Ae^{-km} \quad (8)$$

$$C = 0.095 + 0.04 sin\left[\frac{360}{365}(n - 100)\right] \quad (9)$$

$$I_{BC} = I_B cos\theta \quad (10)$$

$$I_{DC} = CI_B\left(\frac{1 + cos\Sigma}{2}\right) \quad (11)$$

$$I_{RC} = \rho I_B(sin\beta + C)\left(\frac{1 - cos\Sigma}{2}\right) \quad (12)$$

where $m$ is the air mass ratio, $k$ is the collector coefficient of the direct beam, $A$ is the insolation component reaching the earth from the sun, $C$ is the atmospheric diffusion factor, $\rho$ is the ground reflectance, $\Sigma$ is the tilt angle, $L$ is the PV latitude, $\delta$ is the solar declination, $n$ is the number of days, $I_{BC}$ is the direct beam radiation, $I_{DC}$ is the diffuse radiation, and $I_{RC}$ is the reflected radiation.

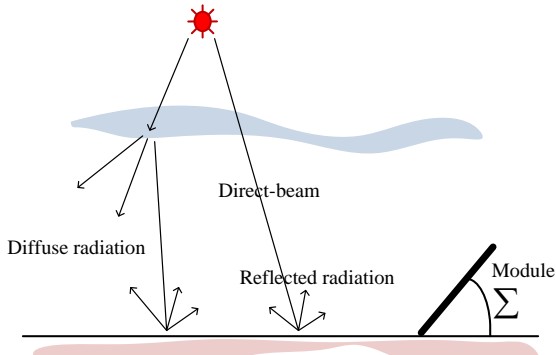

**Figure 5.** Solar radiation striking a collector $I_C$ is a combination of direct beam $I_{BC}$ diffuse $I_{DC}$, and reflected $I_{RC}$ radiation.

Combining the equations for the three components of radiation yields the following for the total rate on a clear day:

$$I_C = I_{BC} + I_{DC} + I_{RC} \quad (13)$$

The calculated radiation can be used to calculate the PV power output. The physics of a PV module can be represented by an equivalent electrical circuit shown in Figure 6 and Equations (14) and (15) [29].

$$I = I_{SC} - I_0\left[exp\left(\frac{V + IR_S}{n \times m\left(\frac{kT}{q}\right)} - 1\right)\right] - \frac{V + IR_S}{R_{SH}} \quad (14)$$

$$I_{SC} = I_{SC(ref)}\left(\frac{I_C}{1000}\right) + J(T - T_{ref}) \quad (15)$$

where $I$ is the output terminal current, $I_0$ is the diode saturation current, $V$ is the terminal voltage of a module, $n$ is the ideal constant of diode, $k$ is the Boltzmann constant, $T$ is the cell temperature, $q$ is the coulomb constant, and $m$ is the number of cells in series in a module.

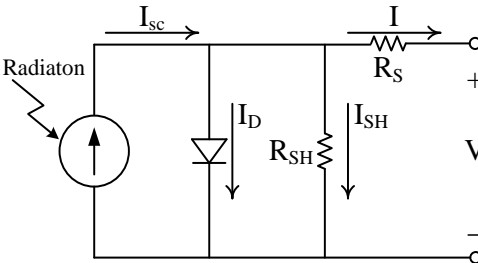

**Figure 6.** PV module equivalent electrical circuit.

The maximum insolation at the Chonnam National University was calculated to predict the photovoltaic output. By applying hourly solar positions, the hourly maximum insolation was calculated on a clear day. The maximum insolation was calculated using the PV installation information listed in Table 1. Figure 7 compares the calculated PV generation with the measured PV generation. The blue dotted line represents the maximum calculated PV power generation. The red line represents the actual measured PV power generation in Chonnam National University.

**Table 1.** PV Installation Information.

| Categories | Values |
| --- | --- |
| Latitude | 35.18° |
| Local Longitude | 126.9° |
| Local Time Meridian | 135° |
| Azimuth Angle | 20° |
| PV Module Tilt Angle | 90° |

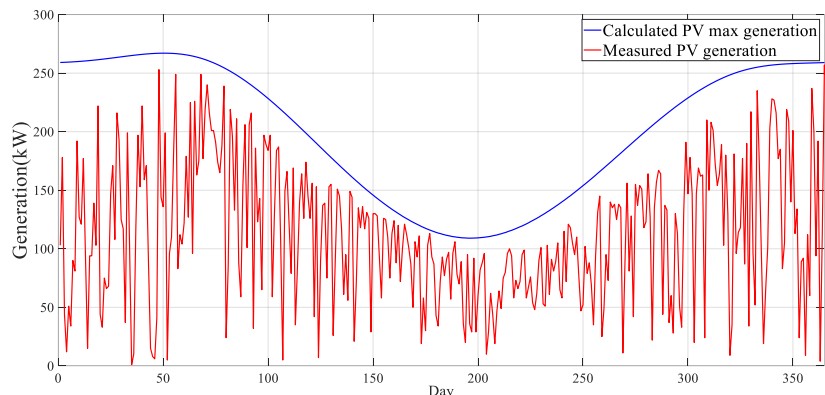

**Figure 7.** Comparison of measured PV generation and calculated PV max generation.

The cloud amount, also known as cloud cover, is the amount of clouds covering the sky. The amount of all clouds floating in the sky is called total cloudiness. It is divided into 11 levels from 0 to 10 based on the amount of clouds. The amount of clouds is expressed as 0 when there is no cloud in the sky and as 10 when the sky is completely covered by clouds. This is expressed in steps as listed in Table 2 [35,36].

**Table 2.** Cloud Coverage [27].

| Classification | Range |
|---|---|
| Clear | 0–2 |
| Partly cloudy | 2–5 |
| Mostly cloudy | 5–8 |
| Cloudy | 8–10 |

The PV generation was determined based on weather factors. The relationship between the weather and PV generation was also analyzed, in relation to the building illustrated in Figure 8. The actual measured output value was compared with the maximum PV power output presented in this paper. By analyzing past data, the average output in each weather type was calculated as shown in Equation (16). The results in Table 3 were derived by analyzing data from 2007 to 2010. If the weather information is known, the PV generation can be estimated as in Table 3.

$$Generation\ rate = \frac{Actual\ output\ of\ PV}{Maximum\ power\ output\ of\ PV} \tag{16}$$

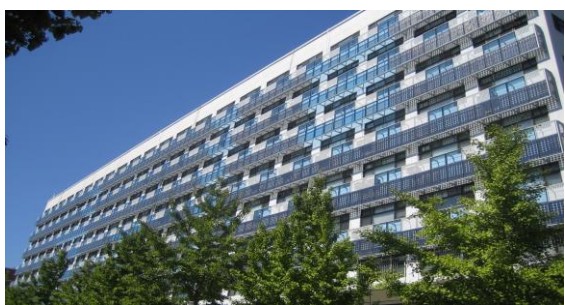

**Figure 8.** 50 kW building integrated PV.

**Table 3.** Generation rates considering 2007–2010 raw data of PV.

| Weather | Generation Rate | Weather | Generation Rate |
|---|---|---|---|
| clear | 0.84 | mostly cloudy, fog | 0.52 |
| clear, fog | 0.73 | mostly cloudy, rain | 0.46 |
| partly cloudy | 0.73 | cloudy | 0.39 |
| partly cloudy, fog | 0.62 | cloudy, rain | 0.24 |
| mostly cloudy | 0.56 | cloudy, fog, rain | 0.24 |

## 4. Operation Method

In this section, the operation method for peak load reduction is proposed. If the weather information and load pattern of the next day are provided, the peak load can be reduced by controlling $P_{SYS}$. In Figure 9, the grid-side power can be seen to level the load. The load forecasting and PV forecasting technology are required for optimal operation of the ESS. In this paper, load prediction is not considered, and the PV maximum output prediction is considered. Loads have pattern characteristics depending on the type (residual, commercial, industrial) and may have similar patterns from day to day. Therefore, the load prediction applied the representative pattern. A brief overview of how to predict the PV generation is presented in Section 3.

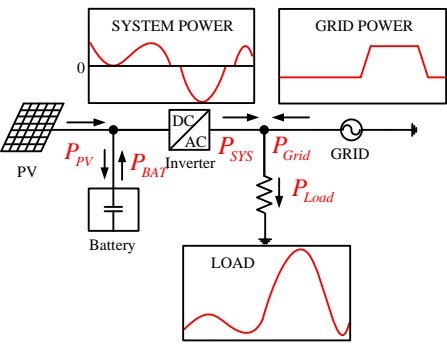

**Figure 9.** Load peak shaving by systems.

An operation method that uses the previously obtained prediction of PV generation and load pattern was proposed. The concept of the proposed method is illustrated in Figure 10. If the hourly demand ($P[h]$) is greater than the system output setpoint ($P_{out}^{set}$), the system power is controlled from the system to the grid. Otherwise, if the hourly load ($P[h]$) is less than the system input setpoint ($P_{in}^{set}$), the system power is controlled from the grid to the system. The given data are the PV generation ($E_{PV}$) during a day, hourly load demand ($P[h]$), full-charge battery energy ($P_{BAT,full}$), initial SoC energy ($E_{BAT,initial}$), and inverter capacity ($P_{SYS,max}$). The system output setpoint ($P_{out}^{set}$) was determined to maximize the reduction in peak load. The system output energy setpoint ($E_{out}^{set}$) was determined to maximize battery energy ($E_{BAT,full}$). The system output power ($P_{out}$) and total output energy ($E_{out}$) were calculated using Equations (17) and (18).

$$P_{out}[h] = max\big((P[h] - P_{out}^{set}), 0\big) \tag{17}$$

$$E_{out} = \sum_{h=1}^{24} P_{out}[h] \tag{18}$$

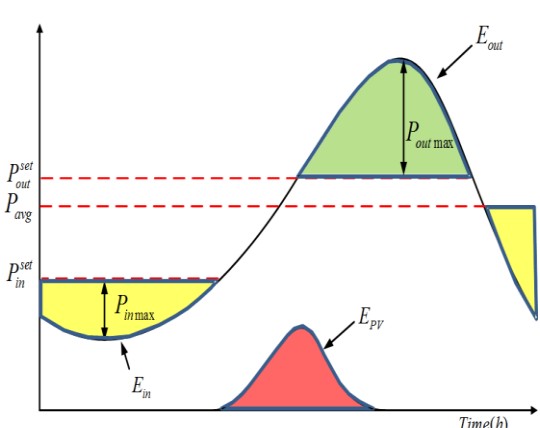

**Figure 10.** Concept of operation algorithm.

The calculated total output energy ($E_{out}$) must be lower than the system output energy setpoint ($E_{out}^{set}$). Otherwise, the system output setpoint ($P_{out}^{set}$), system output power ($P_{out}$), and total output energy ($E_{out}$) were calculated while increasing the system output setpoint ($P_{out}^{set}$). While repeating the previous steps, the final system output setpoint ($P_{out}^{set}$) was calculated.

If the sum of the PV generation ($E_{PV}$) and the initial SoC energy ($E_{BAT,initial}$) is less than the total output energy ($E_{out}$), the system input setpoint ($P_{in}^{set}$) can be set. The system input setpoint ($P_{in}^{set}$) sets the maximum system power ($P_{avg}$) and sum of the minimum load ($P_{min}$) and inverter capacity ($P_{SYS,max}$). The system input power ($P_{in}$) and total input energy ($E_{in}$) were calculated using Equations (19) and (20). $E_{in} + E_{PV} + E_{BAT,initial}$ must be greater than the full-charge battery energy ($E_{BAT,full}$). However, if it is less than the total output

energy ($E_{out}$), the system output energy setpoint ($P_{out}^{set}$) is reset as $E_{in} + E_{PV} + E_{BAT,initial}$. All the previous steps were repeated. The detailed flowchart of the proposed method is shown in Figure 11. First, information related to weather, load patterns, and the battery and inverter ratings are entered. Then, the proposed method performs the parts of Equations (17)–(20) described earlier. In this paper, the normal state of all systems is considered, and over-discharge and overcharge of the BESS are not considered. In the proposed method, over-discharge and overcharge of the BESS can be solved through the constraints of the PCS in BESS, but it is not considered in detail.

$$P_{in}[h] = max\left((P_{in}^{set} - P[h]), 0\right) \tag{19}$$

$$E_{in} = \sum_{h=1}^{24} P_{in}[h] \tag{20}$$

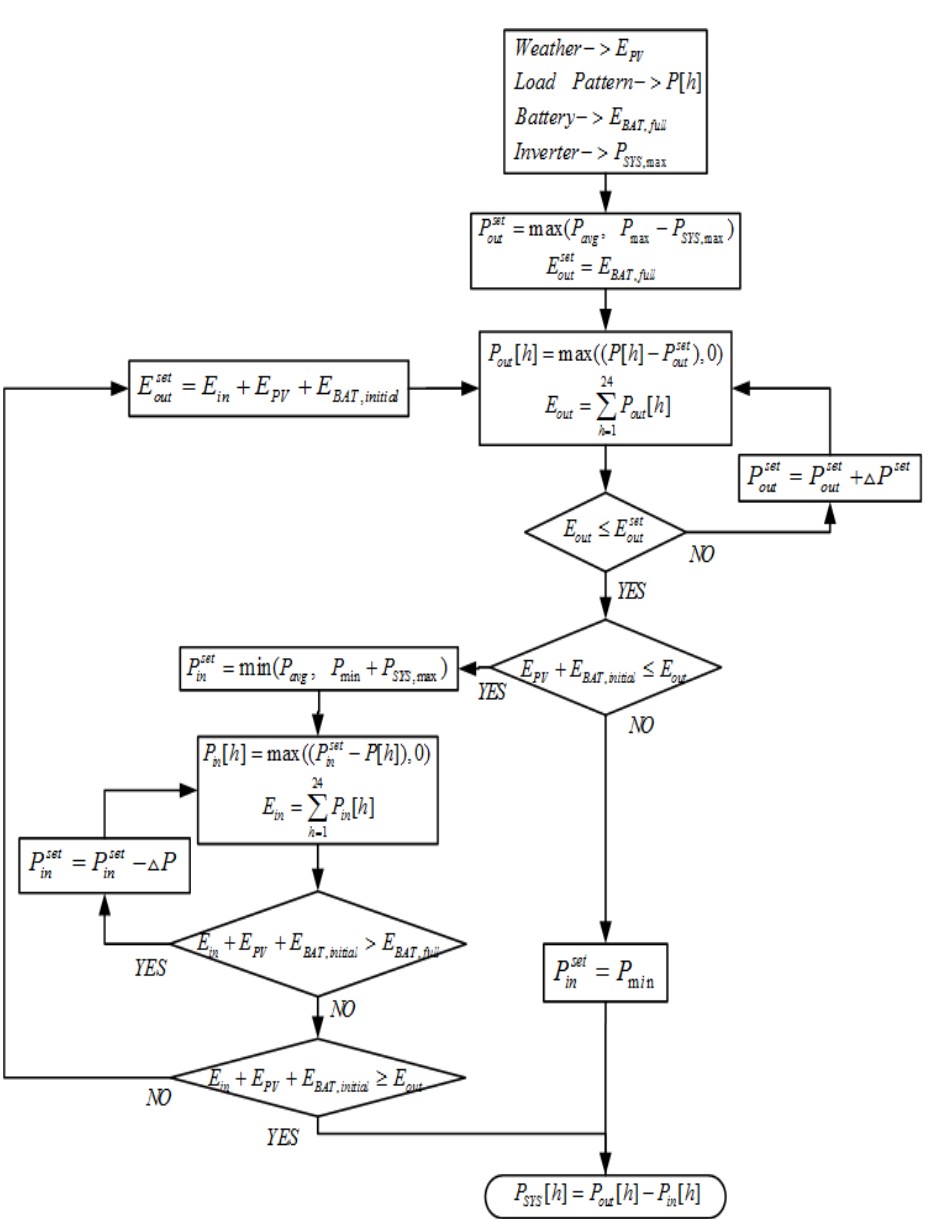

**Figure 11.** Flowchart of the operation method.

## 5. Simulation Studies

The method was implemented using MATLAB to verify the effectiveness of the operation method. Figure 12 shows the residential load patterns. Table 4 lists the system-setting values. Because the PV generation prediction was inaccurate, the initial SoC of the battery sets was 0.2. The initial SoC included the PV generation error. If the predicted PV generation is less than the actual power generation, the initial battery energy can be covered. Otherwise, the additional PV generation energy is fed to the grid. Because the simulation was being conducted to verify the algorithm, the battery operating range was assumed to be 0 to 100%. In actual operation, it can be calculated as a range that can prevent overcharge and over-discharge. The simulation sampling time was 1 h. In this paper, dynamic operation (short period) is not considered and normal operation is considered. However, the operating time may be determined according to the predicted load cycle and PV, and in general, the predicted cycle in the distribution system is performed in units of 15 min, 30 min, and 1 h. The battery capacity is set to a high capacity to maximize the load peak shaving effect of the PV–BESS system. Similarly, the average load was set higher than the PV to maximize the load peak shaving effect.

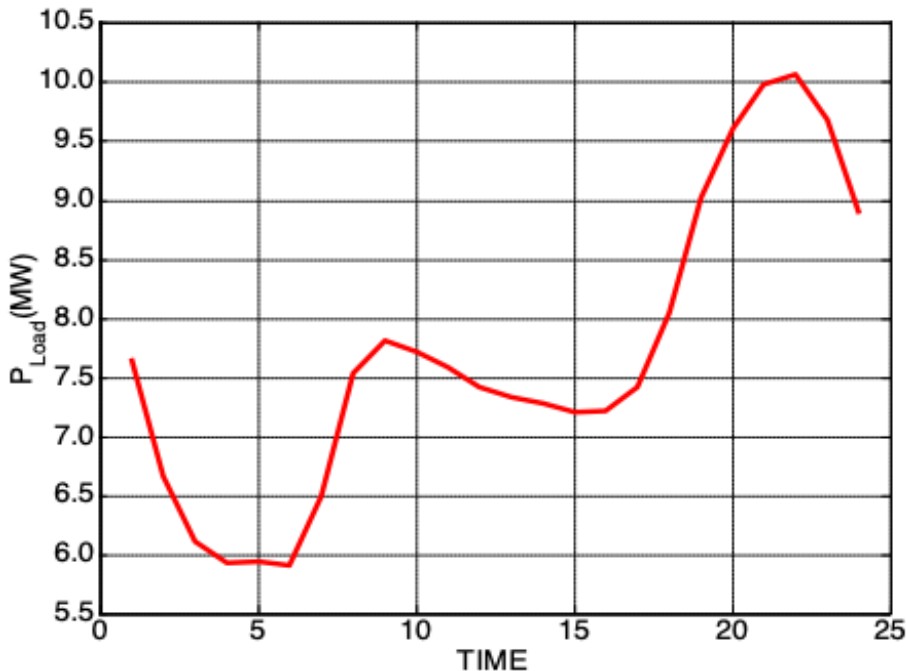

**Figure 12.** Residential load pattern.

**Table 4.** Simulation Setting Value.

| Categories | Setting Values |
| --- | --- |
| Average Load | 7.69 MW |
| Battery Capacity | 12 MWh |
| PV Capacity | 2.5 MWp |
| Inverter Capacity | 2.5 MW |
| Initial SoC (margin) | 0.2 |

Figure 13 shows the results of the system output from the operating algorithm on 12 March 2008. The first graph shows the original load, the second graph shows the PV generation prediction, the third graph shows the grid-side load, and the last graph shows the battery SoC. The weather information on the day was clear and fog (generation rate: 0.73). The results of the simulation were as follows: the system output setpoint, total output

energy, system input setpoint, and total input energy were 7.959 MW, 9.6 MWh, 7.004 MW, and 9.6 MWh, respectively.

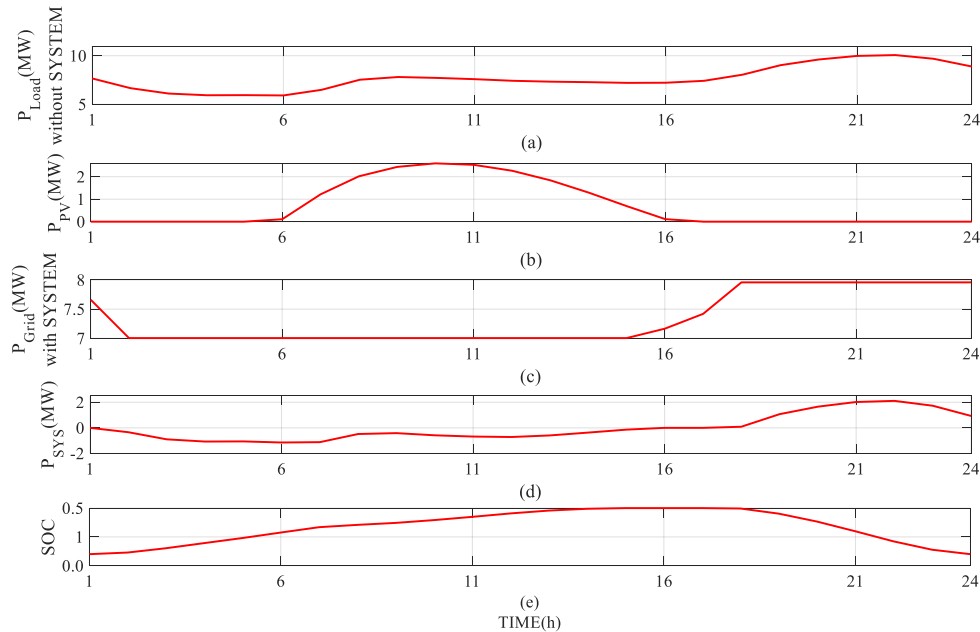

**Figure 13.** 12 March 2008 system output schedule.

Figure 14 shows the results of the system output for 13 March 2008. The weather information included cloudy, fog, and rain (generation rate: 0.24). The results of the simulation were as follows: the system output setpoint, total output energy, system input setpoint, and total input energy were 7.959 MW, 9.6 MWh, 7.397 MW, and 9.6 MWh, respectively. The difference between the two simulations is the input setpoint. As the PV generation rate decreases, the BESS charges more power from the grid.

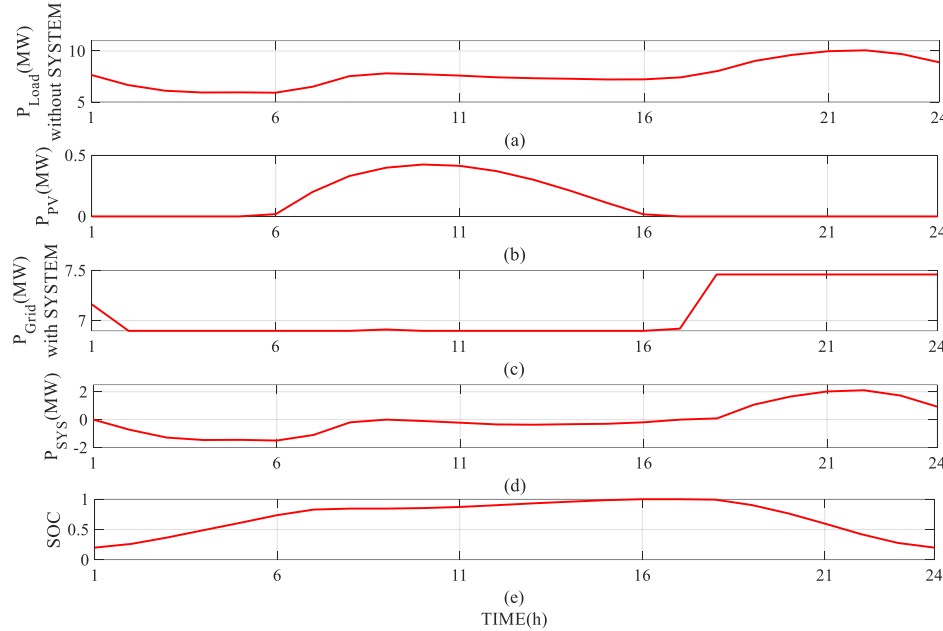

**Figure 14.** 13 March 2008 system output schedule.

To confirm the algorithm results, they were compared with the initial load as shown in Figures 15 and 16. According to the simulation results, it was possible to reduce the power

peak by 30%. Running simulations on annual data is expected to reduce 30% of annual power peaks.

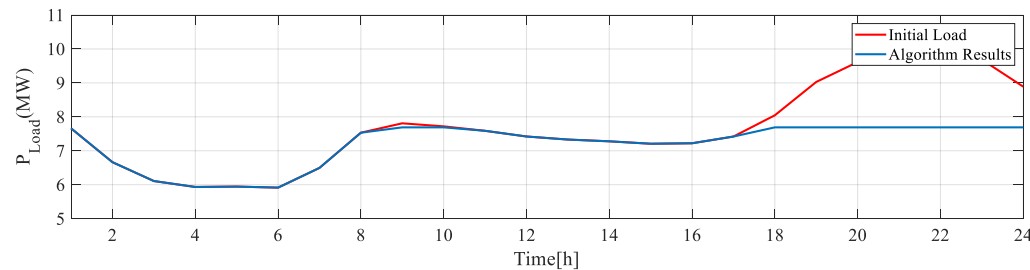

**Figure 15.** Comparison of initial load and algorithm results on 12 March 2008.

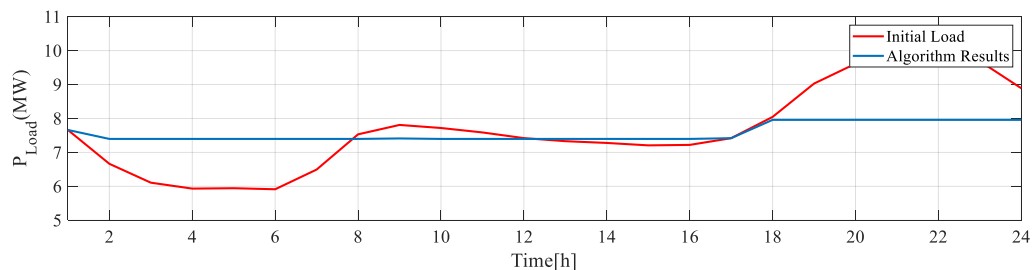

**Figure 16.** Comparison of initial load and algorithm results on 13 March 2008.

## 6. Conclusions

This paper presented an operation method for hybrid PV–battery systems. The method aims to reduce the peak load of the power system by using the estimated load and PV generation of the next day. The PV generation was estimated using historical data analysis, and the load was derived from the load pattern. Using the estimated data, the charge and discharge setpoints of the battery can be determined by considering the initial SoC and capacity of the battery. The effectiveness of the operation method was evaluated using simulations. According to the simulation results, it was possible to reduce the power peak by 30%.

The PV generation output estimation method is a technique that calculates the amount of sunlight and maximum output and estimates the next day's PV output through comparison with the actual output. The proposed method can predict the amount of solar power generation and help power system operation. Moreover, the PV systems can improve operational efficiency and battery-operated algorithms can reduce load peaks. By reducing the peak load of the power system, the power demand imbalance can be resolved. The proposed method has the advantage of increasing power system operating efficiency, which can be further improved by using batteries.

In future work, we will develop an operation algorithm that considers more practical factors, such as the charge/discharge efficiency of the battery system and power loss in the electronic components.

**Author Contributions:** K.-Y.J. prepared the manuscript and implemented the theory and simulations. S.-I.G. supervised the study. All authors have read and agreed to the published version of the manuscript.

**Funding:** This study was supported by the "Regional Innovation Strategy (RIS)" through the National Research Foundation of Korea (NRF) and funded by the Ministry of Education (MOE) (2021RIS-002). This study was supported by a research fund from Honam Univsersity, 2021.

**Data Availability Statement:** Not applicable.

**Conflicts of Interest:** The authors declare no conflict of interest.

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
