# Peer review of "Operation Method of PV–Battery Hybrid Systems for Peak Shaving and Estimation of PV Generation"

_electronics, doi:10.3390/electronics12071608_

Round 1

Reviewer 1 Report

This paper presents an operation method for PV battery hybrid systems by estimating PV generation. Using the PV installation information, the maximum PV generation on a clear day was predicted and compared with historical data. The PV generation was estimated using historical data from 2007 to 2010. The research work reported is interesting in the community. Some suggestions are listed below to improve the manuscript's quality.

1. The manuscript's motivations should be further highlighted in the manuscript, e.g., what problems did the previous works exist? How to solve these problems?

2. The authors must clearly explain the difference(s) between the proposed method and similar works in the introduction.

3. The authors should further highlight the manuscript's innovations and contributions.

4. In line 270,in Table 4. Simulation Setting Value, how to determine these values?

5. In page 6, all acronyms and variables in equations(5)~(12) must be defined in the article.

6. In order to highlight the introduction, some latest references should be added to the paper for improving the reviews part and the connection with the literature. For example, https://doi.org/10.1109/TIM.2020.2983233; https://doi.org/10.3389/fendo.2022.1057089; https://doi.org/10.1016/j.marstruc.2022.103338 and so on.

Reviewer 2 Report

The paper gives new method for maximization of efficiency of system operation by fully utilization of battery. Method was validated trough simulation studies.

-        -  Please explain the differences between measured PV generation and calculated PV max generation in Fig. 7 (pg. 8).

-      -    Regarding operation method described in chapter 4 and given in Fig. 9, can you explain what would be the biggest problems regarding method implementation in real system like you had in yours paper?

-         - Regarding effectiveness of operation method on pg. 11, explain how will you prevent the over-charge or over-discharge in actual operation?

-         - Efectiveness of the operation method was evaluated using simulation, do you expect that in real system application it will be possible to reduce the power peek by 30% ?

Reviewer 3 Report

Comments:

1. The authors claimed that using weather information and load pattern for the next day, the charge and discharge set points of the battery has been determined by considering the initial SoC (State of Charge) and 16 capacity of the battery. However, the method the authors have used for estimation of the PV generation it is not feasible and obsolete. Because there are now many machine learning and deep learning algorithms are available which give better accuracy than the method proposed in this work.

2. The literature review does not contain any recent research work on the proposed work which means this work is no longer valid. 

3. The gap and contribution of the research with adequate discussion is missing.

4. The author claimed that there is one day ahead load forecasting has been implemented. However, in the paper there no such evidence has been observed.

5. There is no comparative study the authors have provided to prove that how the work is better than the recent works.

6. 90% of the references are older than 2015.

7. The data authors have chosen for prediction is 13 years old which is 2010. The reviewer do not think that such old data would be helpful for the current researchers or service providers. Because in last 13 years load pattern and PV irradiation pattern in al over the world has been changed.

Reviewer 4 Report

abstract:

- well written

Intro:

- problem statement not really details. explain more and describe it in comparison with previous research work. 

- reviews more on battery storage rather the overall RE. 

 - Add manuscript outline at the end of this research paper. 

Session 2 and 3:

- okay

Session 4:

- explain in detail of flowchart 11

session 5:

- Sampling time?

- Dynamic operations can be added to the value of the time response.

conclusion:

- add additional results 

Round 2

Reviewer 1 Report

I have appreciated the deep revision of the contents and the present form of this manuscript. All my previous concerns have been accurately addressed. I think that this paper can be accepted.

Reviewer 3 Report

The authors failed to address the comments given by the reviewer. 

Reviewer 4 Report

well-written manuscripts with modifications made from previous comments. Double-check any formatting errors and grammatical errors. 

Round 3

Reviewer 3 Report

The authors need to incorporate a comparative study to prove that how the work is better than the recent works published in between (2018 to 2023).
